# Impact of Primarily Emitted Oxygenated Volatile Organic Compounds on Ozone Formation in the Yangtze River Delta Region

Xun Li<sup>1</sup>, Xuan Li<sup>1</sup>, Rusha Yan<sup>2</sup>, Yaqin Gao<sup>2</sup>, Kangjia Gong<sup>1</sup>, Hongli Wang<sup>2</sup>, Momei Qin<sup>1</sup>, Jianlin Hu<sup>1</sup>, Jingyi Li<sup>1,\*</sup>

<sup>1</sup>Jiangsu Key Laboratory of Atmospheric Environment Monitoring and Pollution Control, Collaborative Innovation Center of Atmospheric Environment and Equipment Technology, School of Environmental Science and Engineering, Nanjing University of Information Science & Technology, Nanjing 210044, China

10 <sup>2</sup>State Environmental Protection Key Laboratory of Formation and Prevention of Urban Air Pollution Complex, Shanghai Academy of Environmental Sciences, Shanghai 200233, China

\*Correspondence to: Jingyi Li (jingyili@nuist.edu.cn)

Abstract. Oxygenated volatile organic compounds (OVOCs) play a crucial role in tropospheric radical chemistry, which in turn enhances atmospheric oxidation capacity and drives the formation of secondary pollutants. However, large uncertainties in their emissions pose challenges to accurately assessing their impacts on regional air quality. This study incorporates updated anthropogenic emission inventories, featuring source-resolved OVOC profiles derived from measurements and literature, into the Community Multiscale Air Quality (CMAQ) model to improve OVOC simulations over the Yangtze River Delta (YRD) region. The model well captured the diurnal and seasonal variations of most OVOCs, especially carbonyls. Primary emissions accounted for 30-70% of total OVOC concentrations, with higher contributions during colder months due to weaker atmospheric oxidation capacity. Hydroperoxyl radicals (HO<sub>2</sub>), the primary oxidant driving NO-to-NO<sub>2</sub> conversion in urban areas, were substantially produced through OVOC photooxidation. Of the HO2 produced by this process, approximately 15-40% originated from directly emitted OVOCs rather than from secondary OVOCs formed via VOC oxidation. Sensitivity analysis further indicated that key emitted OVOCs contributed to ozone formation at levels comparable to traditional VOC precursors. These findings underscore the critical yet often overlooked role of primary OVOC emissions in urban ozone formation, highlighting the need for more comprehensive assessments in regions like the YRD.

# 1 Introduction

Tropospheric ozone (O3) pollution arises from the continuous oxidation of nitric oxide (NO) to nitrogen dioxide (NO2) by hydroperoxyl (HO2) and organic peroxy (RO2) radicals. These radicals are generated through initiation processes, such as the photolysis of oxygenated volatile organic compounds (OVOCs) and the ozonolysis of unsaturated VOCs, and are subsequently recycled via radical chain propagation reactions. Under polluted conditions, OVOC photolysis can become a dominant radical source, substantially enhancing O<sub>3</sub> formation (Yang et al., 2024; Tan et al., 2019b; Xue et al., 2016; Qu et al., 35 2021; Stockwell et al., 2021). Box model simulations constrained by observed OVOC concentrations have significantly improved the simulated HO2, RO2, and hydroxyl (OH) radical levels (Wang et al., 2022b; Wang et al., 2023; Yang et al., 2022), underscoring the critical role of OVOCs in radical budgets. These findings highlight the need for accurate representation of OVOC speciation and concentrations in 40 air quality models to support effective O3 mitigation strategies. Significant discrepancies persist between modeled and observed OVOC concentrations. Box models driven by observed pollutant concentrations, which reflect the chemically aged residuals of reactive species, may misrepresent OVOC variations and thus deviate from observations. As OVOCs are generated through the multi-step oxidation of VOCs, chemical transport model biases are strongly influenced by uncertainties in VOC precursor emissions, which largely depend on the accuracy of activity data and emission factors (Smith et al., 2022). In addition, the chemical production and loss pathways of OVOCs play a critical role. For example, the yield of formaldehyde (HCHO), the simplest aldehyde, is highly sensitive to isoprene chemistry and can directly affect ozone production rates (Marvin et al., 2017). The uptake of small aldehydes and organic acids by deliquesced particles represents an important source of secondary organic aerosols (SOAs). While the SOA formation mechanisms of glyoxal and methylglyoxal have been extensively studied, the contribution of other small aldehydes and organic acids remains poorly quantified due to limited studies and sparse observational data (Gkatzelis et al., 2021). Another important yet often overlooked factor lies in the uncertainties associated with OVOC emissions. Only a limited number of OVOC species are explicitly represented in most emission inventories, partially due to the constraints of traditional detection techniques (Pfannerstill et al., 2023; Wang et al., 2022a). Moreover, VOC and OVOC sources are typically aggregated into broad categories (e.g., industry,

transportation, residential, and power), and their total emissions are generally reported on a monthly or

annual basis. This introduces substantial uncertainties in temporal and spatial allocations, particularly for industrial sources that encompass diverse processes and exhibit sector-specific characteristics (Hu et al., 2025). In addition, emerging sources of OVOC emissions, such as ancillary solvent use associated with vehicles, are not yet incorporated into emission estimation methodologies (Cliff et al., 2023). Collectively, uncertainties in emissions and chemical mechanisms contribute to substantial underestimations of certain OVOCs, such as HCHO, alcohols, ketones, and organic acids, particularly in urban environments dominated by anthropogenic emissions (Luecken et al., 2018; He et al., 2024; Liu et al., 2023; Pfannerstill et al., 2023). In recent years, several studies have developed OVOC emission profiles with improved speciation and quantification accuracy (Wang et al., 2024a; Wang et al., 2022a; Ou et al., 2015; Mo et al., 2016). Nevertheless, differences in emission development methods and the limited representation of source categories lead to substantial variations in reported OVOC and VOC source profiles (Niu et al., 2023), which hinder systematic investigations into the spatiotemporal characteristics of OVOCs and their roles in regional ozone formation.

In this study, an updated emission inventory with refined profiles of OVOCs and their VOC precursors was incorporated into the Community Multiscale Air Quality (CMAQ) model to improve and evaluate OVOC simulations over the Yangtze River Delta (YRD) region in China. Our previous work demonstrates the critical role of OVOC oxidation in enhancing atmospheric oxidation capacity and promoting ozone formation in this region, based on top-down emission adjustments constrained by field observations (Li et al., 2022a). Here, we adopt a process-oriented approach by integrating sector-specific source profiles into the emission inventory to quantify the contributions of primary emissions to ambient OVOC levels in the YRD. We further investigate the respective roles of emitted and secondary OVOCs in ozone and HO<sub>2</sub> production. A subsequent sensitivity analysis is conducted to assess the relative impacts of emitted OVOCs and traditional VOC precursors on ozone mitigation during a pollution episode. These findings advance our understanding of OVOC fate in the atmosphere and their impact on urban ozone chemistry.

### 2 Methodology

# 2.1 Model description

The CMAO model version 5.2, coupled with the SAPRC07tic chemical mechanism and the AERO6 aerosol module, was applied to simulate OVOC distributions in the YRD during March 27-April 30 (EP1) and October 15-November 20 (EP2) of 2019. Several model updates, including heterogeneous reactions of sulfur dioxide (SO2), NO2, glyoxal, and methylglyoxal, were implemented as described in a previous study (Mao et al., 2022). Two nested domains were configured, with the outer domain (d01) covering eastern China and the inner domain (d02) encompassing the YRD. These domains had horizontal resolutions of 12 km × 12 km (127 × 202 grid cells) and 4 km × 4 km (238×268 grid cells), respectively (Fig. S1). Meteorological fields were generated using the Weather Research and Forecasting (WRF) model version 4.2.2, with initial and boundary conditions provided by the fifth generation ECMWF atmospheric reanalysis of the global climate (ERA5). Anthropogenic emissions in d01 were derived from the Multi-resolution Emission Inventory for China (MEIC) version 1.4 (Geng et al., 2024) for mainland China and the Regional Emission inventory in ASia (REAS) version 3.2.1 (Kurokawa and Ohara, 2020) for other Asian countries and regions. For d02, the 2019 YRD emission inventory incorporating updated VOC profiles (2019 YRD inventory) was applied to the YRD, while the MEICv1.4 was used for the remaining regions. Biogenic emissions were generated using the Model of Emissions of Gases and Aerosols from Nature (MEGAN) version 2.1. Open biomass burning emissions were based on the Fire INventory from NCAR (FINN) version 2.5 (Wiedinmyer et al., 2023). The spatial and temporal allocations of anthropogenic and open biomass burning emissions followed previous work (Hu et al., 2016; An et al., 2021). Notably, the temporal allocation did not account for temporary emission control measures, which may partly explain the model biases discussed later. The first five days of each simulation were treated as spin-up and excluded from the analysis. A complete list of all OVOCs and their precursors used in the model is provided in Table S1. The 2019 YRD inventory was developed based on the 2017 version (An et al., 2021), with updated activity data for each source sector. Key improvements included refined VOC speciation, particularly of OVOCs, for major source categories such as transportation (gasoline and diesel vehicles), industrial processes, and residential biomass burning (Gao et al., 2022; Wang et al., 2022a). The chemical composition of VOCs from diesel vehicles, industrial processes, and biomass burning was characterized

based on 160 localized, source-resolved measurements conducted in China. Details of the sampling and analytical procedures can be found in our previous study (Gao et al., 2023). VOC profiles for other sources, such as gasoline vehicles, were updated based on published literature (Wang et al., 2022a; Huang et al., 2024).

As a result of these refinements, OVOCs accounted for 56.8% of total VOC emissions from diesel vehicles (DVE), followed by 44.6% from residential biomass burning (RBB) and 39.1% from industrial processes (INP) (Table S2). Among all OVOCs, carbonyl species showed significant enhancements in these three source categories. Aldehyde contributions nearly doubled in DVE and INP, with a particularly large increase in HCHO. In contrast, the emissions of alcohols, such as methanol (MEOH) and ethanol (ETOH) from INP, decreased. Contributions from other OVOCs, including acetic acid (CCOOH) as well as phenols and cresols (CRES), also increased in INP and DVE. Although the overall OVOC fraction from gasoline vehicle emissions slightly decreased, the chemical composition exhibited a marked change, with increases in carbonyls and decreases in alcohols and esters.

# 125 **2.2 Observation data**

A suite of pollutant and meteorological observations from ground monitoring stations (Fig. S1) was collected for model evaluation. High time-resolution measurements of 77 OVOCs, including 14 aldehydes and ketones, 23 organic acids and esters, 10 furan compounds, and 30 oxygenated aromatic compounds, were recorded at a 10 s interval using a Proton Transfer Reaction-Quadruple interface Time-of-Flight Mass Spectrometer (PTR-QiTOF) at the Shanghai Academy of Environmental Sciences. After removing outliers (values below the background level) and missing data, the remaining data were averaged to hourly means. In addition, C<sub>2</sub>–C<sub>12</sub> hydrocarbons, C<sub>2</sub>–C<sub>5</sub> carbonyls, and C<sub>1</sub>–C<sub>4</sub> alcohols were measured using an online gas chromatography system equipped with a mass spectrometer and a flame ionization detector (GC-MS/FID, TH-300, Wuhan Tianhong Instruments, China) at an hourly resolution. Formaldehyde and peroxyacetyl nitrate (PAN) were measured using a commercial Hantzsch fluorimetry monitor and an online gas chromatography-electron capture detector (GC-ECD), respectively. Details on the instrumentation, analytical procedures, and data quality assurance have been described in previous work (Gao et al., 2022; Liu et al., 2019; Du et al., 2025).

Hourly O<sub>3</sub> concentrations in 14 cities across the YRD were obtained from the China National

Meteorological parameters, including temperature, relative humidity, wind speed, and wind direction, were obtained for four representative cities (Shanghai, Nanjing, Hefei, and Hangzhou) from the China Meteorological Administration (http://data.cma.cn/; last access: 1 November 2024). To evaluate model performance, statistical metrics such as Pearson correlation coefficient (R), mean bias (MB), mean error (ME), normalized mean bias (NMB), normalized mean error (NME), root mean square error (RMSE), and index of agreement (IOA) were calculated and compared against benchmark values where available. Additional details are provided in Text S1.

### 3 Results and Discussion

### 3.1 Characteristics of OVOCs

Among all the observed OVOCs, alcohols were the dominant species in spring (EP1), with ethanol exhibiting the highest hourly median concentrations, ranging from 6 to 12 ppb, which dropped to below 0.5 ppb in fall (EP2) (Figs. 1 and S6). The model exhibited opposite biases, underestimating ethanol in EP1 while overestimating it in EP2. This discrepancy may be attributed to uncertainties in the emission inventory. The monitoring site is located in a typical urban environment surrounded by commercial and residential buildings and busy roads, where emissions from transportation and solvent use have a strong influence (Liu et al., 2019; Liu et al., 2021). One likely underestimated source of alcohols is the use of alcohol-containing solvent products such as windshield washer fluid, particularly from electric vehicles (Cliff et al., 2023). Considering the rapid growth in electric vehicle ownership in China in recent years, this alcohol source will likely become more important. Another potential source of alcohols is the use of household cleaning and personal care products, which may not be fully represented in the current emission inventory (Mo et al., 2021; Wang et al., 2024b). In contrast, the overestimation in EP2 may result from the model's omission of temporary emission restrictions implemented in the period before and during the China International Import Expo 2019 (CIIE 2019, October 27-November 10, 2019) (Wang et al., 2025). Observational data show that ethanol concentrations remained extremely low (

opposite model biases between the two episodes may be partially attributed to the inadequate treatment 170 of temperature-dependent evaporative emissions (Qin et al., 2025), given ethanol's high volatility and the 2.5°C (16.8%) higher average temperature in EP1 compared to EP2 (Fig. S8). Formic acid and acetic acid were the most abundant organic acids during both episodes, but were consistently underestimated by the model, with larger biases in EP2 (Fig. S6 and Table S5). These discrepancies may be partially attributed to uncertainties in the emission inventory. Acetic acid showed 175 strong correlations with ethyl acetate, methyl ethyl ketone (MEK), and toluene (Pearson coefficients of 0.75-0.85, p 

since biogenic sources of acetone are important (Lyu et al., 2024; Hu et al., 2013), the larger discrepancy observed in the warm season may also be due to the model's inadequate representation of urban green vegetation (Maison et al., 2024; Ma et al., 2022).

Notably, biases in OVOC concentrations are influenced by those of their precursors. For example, model biases in methacrolein (MACR) and methyl vinyl ketone (MVK) exhibited similar temporal patterns to those of isoprene (ISOP), reflecting their common origin via isoprene oxidation. The model reasonably reproduced key OVOC precursors, including reactive alkanes (ALK3/ALK4), aromatic hydrocarbons, and alkenes (e.g., ethylene and propylene), particularly during EP1. This suggests that biases related to secondary OVOC production were less pronounced in EP1 than in EP2, consistent with the larger discrepancies observed in EP2 (Table S5). The simultaneous overestimation of VOC precursors and underestimation of OVOCs also indicates that uncertainties in their photochemical mechanisms may contribute to the model biases in OVOC concentrations.

At the regional scale, average OVOC concentrations across the YRD ranged from 5 to 25 ppb in both seasons, with elevated levels observed in the central and eastern areas (Figs. 2a and 2d). HCHO, MEOH, ACET, and acetaldehyde (CCHO) were the dominant species (Fig. S10). Most OVOCs exhibited higher concentrations in EP2 than in EP1, except for MEOH, likely due to its stronger biogenic contribution during the warm season, particularly in the southern and southwestern YRD. On average, the model predicted that approximately 30–70% of total OVOCs originated from primary emissions (primary OVOCs) rather than from VOC oxidation (secondary OVOCs). The contributions of primary OVOCs show significant seasonal and spatial variations. During the warm season, secondary OVOCs dominated across the entire domain, while elevated emission contributions were observed in hotspots along the Yangtze River and in Shanghai (Fig. 2b). In contrast, during the cold season, primary OVOCs played a larger role in Jiangsu Province as well as in Shanghai and Hangzhou (Fig. 2e), consistent with our previous findings for October 2017 in the YRD(Li et al., 2022a).

Figure 1. Diurnal variations of OVOCs and their precursors in Shanghai, April 2019 (ppb). Red and black circles indicate the median observations and model predictions, respectively; shaded areas and black error bars represent the interquartile ranges (25th-75th percentiles) of observations and model predictions, respectively.

CRES accounted for the highest OVOC emission rate across the YRD (22%), followed by HCHO (16%), ACET (14.5%), CCOOH (12.5%), and ETOH (9.5%) (Table S6). High emission contributions to ambient CRES and CCOOH were also observed, followed by ACET (Fig. S11). Primary emissions accounted for approximately 20–60% of ambient HCHO, with maxima reaching up to 76% in EP1 and 88% in EP2 in Jiangsu Province. The model also estimated a substantial fraction of biacetyl (BACL) originating from primary emissions (more than 60%), particularly in Jiangsu and Anhui Provinces, as well as Shanghai.

Figure 2. Episode-averaged total OVOC concentrations (a, d), primary emission contributions to total OVOCs (b, e), and daily maximum 8-hour average ozone (MDA8 O<sub>3</sub>) concentrations (c, f) during the two episodes.

The spatial distributions of OVOCs and ozone were closely aligned, with OVOC hotspots coinciding with regions of elevated daily maximum 8-hour average ozone (MDA8 O<sub>3</sub>) concentrations (Figs. 2c and 2f). Further analysis revealed that OVOC photooxidation constituted a significant source of HO<sub>2</sub> radicals in the YRD, serving as a key driver of NO-NO<sub>2</sub> cycling and thereby influencing ozone formation.

# 3.2 Role of OVOCs in Ozone Formation

To investigate the relationship between OVOCs and  $O_3$  formation at the monitoring site in Shanghai, a "clean period" (April 1–4) and a "pollution period" (April 5–7) were selected. During the clean period, observed MDA8  $O_3$  concentrations (81.4–139.4  $\mu$ g/m³) complied with China's Ambient Air Quality Standards (GB 3095-2012) threshold of 160  $\mu$ g/m³. In contrast, MDA8  $O_3$  levels during the pollution period ranged from 161.3  $\mu$ g/m³ to 189.1  $\mu$ g/m³, exceeding this standard. The model showed good agreement with observations, successfully capturing temporal variations of both MDA8  $O_3$  and OVOC concentrations during these periods, demonstrating its capability to represent OVOC sources and sinks as well as their role in ozone formation.

The ozone production rate  $(P(O_3), ppb/h)$ , defined as the rate at which ozone is produced through the conversion of NO to NO<sub>2</sub> by peroxy radicals, was calculated as follows:

$$P(O_3) = k_0[HO_2][NO] + \sum_i k_i [RO_2^i][NO],$$
 (1)

where k<sub>0</sub> and k<sub>i</sub> are the rate constants for the reactions of NO with HO<sub>2</sub> and RO<sub>2</sub> radicals (unit: molec<sup>-1</sup> cm<sup>3</sup> s<sup>-1</sup>), respectively, and [HO<sub>2</sub>], [RO<sub>2</sub>], and [NO] denote their respective concentrations (unit: ppb). In 255 RO<sub>2</sub> + NO reactions, alkyl nitrates can form as byproducts, with yields (α) depending on the specific RO<sub>2</sub> species. Accordingly, a NO<sub>2</sub> yield of (1-α) was applied to the corresponding RO<sub>2</sub> + NO reactions. The NO<sub>2</sub> molar yields for reactions producing alkyl nitrate coproducts in the current model are listed in Table S7. The analysis of P(O<sub>3</sub>) and radical chemistry is focused on the daytime period (9:00 - 17:00 local time). As shown in Figs. 3a and 3b, the daytime average P(O<sub>3</sub>) during the clean period was approximately half 260 that under the polluted conditions, at 5.89 ppb/h and 12.2 ppb/h, respectively. HO<sub>2</sub>-driven oxidation accounted for 51-59% of daytime P(O<sub>3</sub>) during the clean period, increasing slightly to 50-61% under the polluted conditions. Ozone production via the HO<sub>2</sub> pathway peaked around noon, coinciding with the highest P(O<sub>3</sub>), and declined gradually to approximately 50% later in the day. These results indicate that 265 the HO<sub>2</sub> pathway consistently dominates ozone production in Shanghai, regardless of pollution levels. The dominant role of HO<sub>2</sub>-driven oxidation in ozone production was further confirmed in other urban areas of the YRD. During both EP1 and EP2, HO2 radicals contributed approximately 55-80% of total ozone production in regions with elevated P(O<sub>3</sub>) (Figs. 4 and S12). These areas, including Shanghai, are heavily influenced by anthropogenic emissions, where HO2 concentrations exceeded those of RO2 270 radicals (Fig. S12). In contrast, RO2 radicals dominated ozone production in the southern and southwestern parts of the domain, where high biogenic VOC emissions and relatively low NOx levels prevail. The enhanced role of RO2 radicals in these regions is attributed to their higher concentrations and comparable reactivities to HO2 radicals in converting NO to NO2. To understand the mechanism of daytime HO2 radical production, source contributions from the primary HO<sub>2</sub> generated by OVOCs and VOCs, RO<sub>2</sub> reactions, and other pathways of inorganic species in Shanghai were investigated (Fig. 3c). It should be noted that HO<sub>2</sub> produced directly during OVOC and VOC oxidation, excluding formation via RO2 chain reactions, is defined as "primary" HO2, whereas HO2 formed through reactions without radical reactants, such as photolysis and O<sub>3</sub>-initiated reactions, is referred to as "newly generated" HO2. The contribution of HNO4 was excluded, as NO2 and HO2 can 280 react rapidly to reproduce HNO<sub>4</sub>. Under the clean conditions, the primary HO<sub>2</sub> production through VOC and OVOC photooxidation dominated among all pathways, accounting for 30.3% and 29.9% of the total, respectively. Reactions involving inorganic species, such as CO + OH, accounted for approximately 22.0%

of total  $HO_2$  production. Another important source was the reactions of  $RO_2$  with NO and  $RO_2$  radicals (17.9%), with the  $RO_2$  + NO pathway being dominant. In the  $HO_2$  pool, approximately 16.6% was newly generated, with OVOC photolysis being the dominant source (94.1%). Similar findings have been reported in other urban areas, where OVOC photolysis accounted for over 80% of newly generated  $HO_2$  radicals, highlighting its importance in  $NO_x$  cycling and ozone production(Qu et al., 2021; Xue et al., 2016; Yang et al., 2018).

Figure 3. Diurnal variations of ozone production rate (a, b) and source contributions to daytime HO<sub>2</sub> production (c-e) under clean and polluted conditions in Shanghai. In panels (a) and (b), red and black circles represent the median P(O<sub>3</sub>) and the contribution of the HO<sub>2</sub> + NO pathway, respectively. The shaded areas and black error bars denote the interquartile ranges. In panel (c), "PVOC" denotes primary HO<sub>2</sub> production from VOC precursors of OVOCs, while "OVOC\_pri" and "OVOC\_sec" denote primary HO<sub>2</sub> production from emitted and secondary generated OVOCs, respectively. "polluted<sub>new</sub>" ("clean<sub>new</sub>") and "polluted" ("clean") refer to newly generated HO<sub>2</sub> and the total HO<sub>2</sub> production, respectively, under the polluted (clean) conditions. Panels (d) and (e) show the fractional contributions of individual OVOCs to total (outer circle) and newly generated (inner circle) primary HO<sub>2</sub> production.

Under the polluted conditions, contributions from inorganic species increased to about one-third of the total HO<sub>2</sub> production, mainly due to elevated CO levels. The primary HO<sub>2</sub> generated by OVOC oxidation became the largest organic source (24.9%), followed by the RO<sub>2</sub> pathway (21.9%). In addition, the fraction of newly generated HO<sub>2</sub> decreased to 9.86%, suggesting an enhanced radical propagation effect and higher atmospheric oxidation capacity under the polluted conditions. As a result, the contribution of

secondary-generated OVOCs to their total primary HO<sub>2</sub> production rose from 71% to 82%. Overall, OVOC photooxidation plays an important role in the HO<sub>2</sub> production across pollution conditions.

Among all OVOCs, HCHO was the most significant contributor to HO<sub>2</sub> (~60%) under both clean and polluted conditions. Under the clean conditions, CRES was the second-largest contributor, followed by photoreactive monounsaturated dicarbonyls from aromatic fragmentation (AFG1) and ethanol. Under the polluted conditions, reactive ketones (PRD2) became the second-largest contributor, followed by AFG1. These species were mainly produced from VOC oxidation. For newly generated HO<sub>2</sub> radicals, glyoxal and methylglyoxal also contributed substantially. These findings align with a modeling study in Beijing, which showed that HCHO, MGLY, and aromatic oxidation products played an important role in HO<sub>2</sub> production(Qu et al., 2021).

Across the YRD, primary HO<sub>2</sub> produced by OVOC oxidation contributed 20–40% of the total HO<sub>2</sub> when the HNO<sub>4</sub> pathway was excluded (Figs. 4b and 4e), exceeding contributions from both VOC precursor oxidation and the RO<sub>2</sub> pathway (Fig. S13). Due to high CO emissions, the CO + OH reaction accounted for 20–50% of the total HO<sub>2</sub> production, with a more pronounced contribution in the northern YRD (Figs. S13c and S13f). OVOC photolysis can become the dominant source of the newly generated HO<sub>2</sub> radicals, contributing up to 98% regionally. Primary emitted OVOCs made substantial contributions to HO<sub>2</sub> production in Jiangsu Province, particularly near Zhenjiang and northern Jiangsu, as well as in Shanghai and Huzhou (Figs. 4c and 4f). Among the total and newly generated primary HO<sub>2</sub> from OVOC oxidation, emitted OVOCs contributed 15–40% and 10–25% (Fig. S14), respectively, highlighting their important role in driving ozone production in the YRD.

Figure 4. Daytime-averaged contributions of the HO<sub>2</sub> + NO pathway to ozone production rates (a, d), contributions of primary HO<sub>2</sub> production from OVOC to total HO<sub>2</sub> radical production (b, e), and contributions from emitted OVOCs (c, f) during the two episodes.

### 3.3 Sensitivity of Ozone Formation to OVOCs and Precursors

Given that a substantial fraction of OVOCs originates from primary emissions, the sensitivity of ozone formation to anthropogenic emissions of OVOCs and their precursors was systematically examined using the CMAQ model coupled with the High-Order Decoupled Direct Method (HDDM). The model configuration followed Li et al(Li et al., 2022b), with the same inputs described in Section 2.1. The ozone responses to the complete removal of various VOCs ( $\Delta$ O<sub>3</sub>) in Shanghai, along with their normalized contributions relative to total anthropogenic emissions across the domain ( $\Delta$ O<sub>3</sub>/ $\Delta$ E<sub>i</sub>) under different polluted conditions, are discussed below.

ppb and 1.01 ppb to ozone formation under clean and polluted conditions, respectively (Fig. 5a). These ozone changes were comparable to those of most VOC precursors except ETHE. BACL was also a key OVOC precursor, contributing 0.35 ppb and 0.50 ppb of ozone under clean and polluted conditions, respectively. BACL undergoes photolysis to produce acetyl peroxy radicals (CH<sub>3</sub>CO<sub>3</sub>), which facilitate NO-to-NO<sub>2</sub> conversion at a rate significantly higher than most other peroxy radicals represented in the model. On a per-unit-mass basis, OVOCs demonstrated substantially higher ozone formation efficiencies than other VOC precursors. Notably, BACL emerged as the most potential contributor in Shanghai, generating  $7.78 \times 10^{-4}$  ppb and  $1.49 \times 10^{-3}$  ppb of ozone per gram emitted under clean and polluted conditions, respectively. HCHO, MVK, MACR, and MGLY also exhibited relatively high ozone formation efficiencies, contributing  $(1.57-3.77) \times 10^{-4}$  ppb per gram emitted across both conditions. Similar sensitivities of O<sub>3</sub> formation to OVOC and VOC precursor emission controls were also observed across the YRD. Among all OVOCs, HCHO demonstrated the most pronounced impact, with emission reductions leading to ozone decreases of up to 2.0 ppb in Jiangsu Province (Fig. S15). This effect was comparable to that of aromatic hydrocarbons (ARO2MN and TOLU) and reactive alkenes (OLE2), and exceeded the influence of most other VOC precursors, except ETHE and propylene (PROP). BACL also exhibited considerable effects, causing up to 1.0 ppb decreases in O<sub>3</sub>, similar to reactive alkanes (ALK3-5), alkenes (OLE1), and most aromatic hydrocarbons. Notably, not all OVOC emission reductions yielded effective ozone mitigation; certain compounds, such as CCHO and CRES, exhibited nonlinear

responses. In some cases, reducing their emissions slightly increased ozone concentrations, likely due to complex interactions with other reactive species. Nonetheless, controlling emissions of OVOCs, particularly HCHO and BACL, represents an efficient strategy for reducing O<sub>3</sub> in the YRD.

Figure 5. Sensitivity of daytime average ozone to individual OVOCs and VOC precursors (a), and emission-normalized ozone changes (b) in Shanghai during the clean and polluted periods.

## 4 Conclusions

OVOCs are key precursors of peroxy radicals that drive NO-to-NO<sub>2</sub> conversion, thereby facilitating O<sub>3</sub> accumulation in the troposphere. Model biases in simulating OVOCs are often attributed to uncertainties in VOC precursor emissions and the photochemical mechanisms used in chemical transport models. However, we emphasize that primary anthropogenic emissions constitute significant yet often underrepresented sources of ambient OVOCs, particularly in urban environments where industrial activities, solvent use, and fuel combustion dominate. High local emission contributions to ambient OVOCs (9–53% for carbonyl compounds and 50–87% for alcohols) have also been reported in other Chinese cities based on field measurements (Wu et al., 2020; Huang et al., 2020). Certain OVOCs, such as formaldehyde, acetone, and alcohols, are highly volatile, and their emission rates are therefore expected to exhibit strong temperature dependence. This inference is supported by direct measurements

significantly higher proportions of OVOCs in total VOC emissions from car painting, waste transfer station, and laundry sources in summer than in winter (Niu et al., 2021). With ongoing climate warming and more frequent heatwave events, emissions of these OVOC species are expected to increase, potentially enhancing atmospheric oxidation capacity and accelerating secondary pollutant formation. The impacts of such temperature-driven increases in OVOC emissions remain insufficiently understood. HO<sub>2</sub> radicals dominated ozone formation in urban areas of the YRD, contributing 50-60% of the daytime ozone production rate under both clean and polluted conditions in Shanghai, and ranging from 55% to 80% across the region during both warm and cold seasons. This is consistent with findings from other urban environments heavily influenced by anthropogenic emissions (Zhu et al., 2024; Tan et al., 2019a; Hu et al., 2023). Therefore, emission controls targeting major HO<sub>2</sub> radical contributors could be an effective strategy for ozone mitigation in urban regions like the YRD. Traditionally, the significance of OVOCs in the free radical budget has been mainly recognized through their formation as secondary products from VOC precursors. However, we demonstrated that primary HO2 produced via OVOC photooxidation accounted for 20-40% of total HO<sub>2</sub> production, with 15-40% of this attributed to emitted OVOCs. As a result, key OVOC species (e.g., HCHO) can compete with their precursors in generating HO<sub>2</sub> radicals and influence ozone formation. In addition, the contributions of OVOCs to RO<sub>2</sub> radicals, such as BACL, also promote ozone production and should not be overlooked. Several uncertainties remain in the current study. Improvements are needed in representing multiphase chemistry, especially in detailed VOC speciation, multistep oxidation reactions, and the heterogeneous production and/or loss of small aldehydes and carboxylic acids. Additionally, emission profiles of organic acids and alcohols require further refinement, along with improved characterization of the temperature dependence of evaporative emissions. Previous studies have shown that alcohols contribute substantially to total OH reactivity and ozone formation, with their impacts intensifying at higher temperatures (Pfannerstill et al., 2023; Luecken et al., 2018; Pusede et al., 2014); however, large discrepancies between modeled and observed concentrations of these compounds were identified in the current study. Moreover, limited observational data confined the analysis to spring and fall. Future studies should extend to all seasons and other regions, focusing on OVOC emission contributions to ambient levels and the sensitivity of ozone formation to OVOC emission reductions, with improved representation of OVOC emissions, speciation, and multiphase chemistry.

of 13 volatile emission sources during a sampling campaign in central China, which revealed

https://doi.org/10.5194/egusphere-2025-4919 Preprint. Discussion started: 13 November 2025

© Author(s) 2025. CC BY 4.0 License.

EGUsphere Preprint repository

Code and data availability

The CMAQ source code can be downloaded at https://github.com/USEPA/CMAQ. The input files to

CMAQ, WRF, and MEGAN, and meteorology and ozone observation data can be downloaded from the

405 links and cited references given in the methods section. VOC observation data at Shanghai and Python

scripts for processing data and generating figures are available upon reasonable request from the

corresponding author.

**Author contributions** 

J.L. designed the research. X.L. and X.L. conducted the simulations and analyzed the data. R.Y., Y.G.,

and H.W. provided the 2019YRD emissions and ground observation data of VOCs at Shanghai. X.L.

drafted the main text. All authors contributed to interpreting the results and editing the manuscript.

**Competing interests** 

The authors declare that they have no conflict of interest.

Acknowledgements

This work was financially supported by the National Key R&D Program of China (grant no.

2022YFE0136200; 2023YFC3706105) and the National Natural Science Foundation of China (grant no.

42077199).

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
