# Peer review of "Impact of Primarily Emitted Oxygenated Volatile Organic Compounds on Ozone Formation in the Yangtze River Delta Region"

_EGUsphere, 2025_

## Referee Comment (RC1)

General comments:

OVOCs play a central role in tropospheric radical chemistry by enhancing the atmospheric oxidizing capacity and promoting the formation of secondary pollutants. This manuscript investigates the role of primary OVOCs in ozone formation over the YRD region by implementing updated, source-resolved OVOC emission profiles into CMAQ and combining model evaluation with process-based and sensitivity analyses. The topic is timely and relevant to the ACP readership, given the growing attention to VOC/OVOC inventories, VCP-related emissions, and radical budgets in polluted regions. However, several issues need to be addressed before the manuscript can be considered for publication in ACP.

Specific comments

1. Line 19, The original statement "The model well captured the diurnal and seasonal variations of most OVOCs". Please clarify whether the model captures the variations in OVOC emission fluxes or in OVOC concentrations. In addition, support the statement more quantitatively (e.g. by quoting ranges of R, NMB, NME for key OVOC classes) rather than relying mainly on qualitative descriptions.

2. Throughout the paper, "primary OVOCs", "emitted OVOCs" and "directly emitted OVOCs" are used. For clarity and consistency (also with the paper title), I recommend adopting "primary OVOCs" as the main term and defining it explicitly in Sect. 2 (e.g. primary = directly emitted from anthropogenic/biogenic sources; secondary = produced via VOC oxidation). Please ensure that the terminology is used consistently in the text, figures, and captions.

3. Section 3.1, This section currently mixes the description of observed OVOC concentration characteristics with the discussion of model simulation results, which makes the content rather long and somewhat fragmented. I suggest splitting it into two subsections, separating the presentation of observations from the evaluation of model performance.

4. Section 3.1, For some species (e.g. acids, alcohol) the discussion emphasizes underestimation or overestimation, but quantitative information on observed mean/median concentrations is limited. It would be helpful to briefly summarize typical concentration ranges for the main OVOC classes in the main text or in a supplementary table, to give readers a clearer sense of magnitude.

5. Lines 213-215: Is there any evidence or supporting analysis to substantiate the conclusion that the biogenic contribution is higher during the warm season?

6. Section 3.2, The authors acknowledge missing or simplified pathways for the production and loss of small aldehydes and organic acids. Given the importance of these processes for formic and acetic acid biases, please expand slightly on which multiphase/heterogeneous reactions are included (e.g. glyoxal/methylglyoxal uptake, aqueous-phase oxidation) and which are missing, and discuss how these omissions could affect both OVOC budgets and the derived role of primary emissions.

7. Section 3.2, The terminology for HO2 sources (primary from VOC/OVOC oxidation, "newly generated" from photolysis and O₃ reactions, etc.) is central to

the conclusions. I recommend adding a concise schematic or a table defining each category before Fig. 3 and clarifying whether "primary" HO2 excludes contributions from RO2 to HO2 recycling. This will help readers interpret Fig. 3c-e and the subsequent regional analysis.

Technical comments
1. Lines 41-43, please insert relevant reference for the box models results.
2. Line 149, I suggest adding the word "concentration" to the section title to more clearly reflect the main focus of this subsection.